# Nuclear Envelope Proteins Modulating the Heterochromatin Formation and Functions in Fission Yeast

**DOI:** 10.3390/cells9081908

**Published:** 2020-08-16

**Authors:** Yasuhiro Hirano, Haruhiko Asakawa, Takeshi Sakuno, Tokuko Haraguchi, Yasushi Hiraoka

**Affiliations:** Graduate School of Frontier Biosciences, Osaka University, 1-3, Yamadaoka, Suita 565-0871, Japan; yhira@fbs.osaka-u.ac.jp (Y.H.); askw@fbs.osaka-u.ac.jp (H.A.); sakuno@fbs.osaka-u.ac.jp (T.S.)

**Keywords:** nuclear envelope, heterochromatin, centromere: telomere, mating type locus, knob, Lem2, LEM domain protein, nuclear pore complex, nucleoporin

## Abstract

The nuclear envelope (NE) consists of the inner and outer nuclear membranes (INM and ONM), and the nuclear pore complex (NPC), which penetrates the double membrane. ONM continues with the endoplasmic reticulum (ER). INM and NPC can interact with chromatin to regulate the genetic activities of the chromosome. Studies in the fission yeast *Schizosaccharomyces pombe* have contributed to understanding the molecular mechanisms underlying heterochromatin formation by the RNAi-mediated and histone deacetylase machineries. Recent studies have demonstrated that NE proteins modulate heterochromatin formation and functions through interactions with heterochromatic regions, including the pericentromeric and the sub-telomeric regions. In this review, we first introduce the molecular mechanisms underlying the heterochromatin formation and functions in fission yeast, and then summarize the NE proteins that play a role in anchoring heterochromatic regions and in modulating heterochromatin formation and functions, highlighting roles for a conserved INM protein, Lem2.

## 1. Introduction

The nuclear envelope (NE) is a membrane structure that surrounds chromosomes and plays a role in providing an appropriate physicochemical environment for chromosomes to modulate their genetic activities. The NE is composed of three components: The double-membrane, the nuclear pore complex (NPC), and the nuclear lamina. The double membrane is composed of an inner nuclear membrane (INM) and an outer nuclear membrane (ONM). ONM is continuous with the endoplasmic reticulum (ER), and thus contains integral membrane proteins common to those of the ER membrane. In contrast, the INM contains various NE-specific integral membrane proteins. The majority of the proteins interact with the nuclear lamina, chromatin, and other NE proteins, and thus can be involved in various chromosomal processes such as heterochromatin formation. The NPC exists as a structure piercing through the INM and ONM. The NPC is a large protein complex with 8-fold rotational symmetry and acts as a gateway for nucleocytoplasmic transport. The nuclear lamina exists beneath the INM, and is known to play an important role in the structure and function of chromosomes by interacting with the chromosome region named lamina-associated domain (LAD) [1]. However, the nuclear lamina exists only in metazoans, including humans, but not in unicellular organisms such as yeasts, and plants [2,3].

A number of NE proteins have been identified in mammalian cells. One of the classical NE proteins is lamin B receptor (LBR), which was first identified as an INM protein from turkey erythrocyte cells [4]. LBR is known to play a role in attaching chromatin to NE and to form heterochromatin underneath the NE in mammalian cells [5,6,7]. LBR exists in various species of metazoans, but does not exist in those outside of metazoan. Another classic NE protein is the LEM domain protein, which was named as an acronym for three proteins: Lamina-associated polypeptide 2 (Lap2), emerin, and MAN1 because the domains in the N-terminal region (approximately 40 amino acid residues) of these proteins are similar [8,9,10]. LEM domain proteins include Lem2, several splicing variants of Lap2, and others, in addition to the three proteins (Lap2, emerin and MAN1) described above [8,9,10]. LEM domain proteins have been reported to be involved in various cellular processes, including retroviral infection, cell cycle control, NE assembly, chromatin organization, and heterochromatin formation [11,12]. Among the LEM domain proteins, Lem2 is highly conserved among various species, from *Tetrahymena* (belonging to SAR) to yeasts and humans (belonging to Opisthokonta) [13], suggesting that it may have conserved functions on the chromosome. In addition to these NE proteins, over 600 NE proteins have been identified in mammalian cells [14], and their cellular functions remain largely unknown.

The NPC is composed of multiple sets of approximately 30 different proteins, called nucleoporins (Nups), which are generally conserved across species. Nups are classified into three groups based on their structural and functional features: Phenylalanine-glycine (FG) repeat Nups, transmembrane Nups, and scaffold Nups. FG repeat Nups are involved in nucleocytoplasmic transport through pores. Transmembrane Nups have transmembrane domains and attach the NPCs to the NE. Scaffold Nups form two ring structures: Inner ring and outer ring, which serve as the NPC structural core [15,16,17,18] and associate with the membrane through interactions with the transmembrane Nups [19,20,21]. Although these NPC structures and most Nups are generally conserved among eukaryotes [22,23,24,25,26,27,28,29], some differences have been found among the species [23,30]. Recently, it has been reported that some of the Nups play a critical role in attaching heterochromatin to the NE [31,32].

Fission yeast *Schizosaccharomyces pombe* is a model organism that has advantages in studies of nuclear membrane proteins as it has a relatively small number of nuclear membrane proteins, and genetic analysis makes it easier to investigate their functions. Recent studies in *S. pombe* demonstrated that some NE and NPC proteins play a role in modulating heterochromatin formation [31,32,33,34,35]. In this review, we highlight the proteins that modulate heterochromatin formation and functions in *S. pombe.* Because these proteins are highly conserved among a wide range of eukaryotes, findings from studies of *S. pombe* will provide general insights into heterochromatin formation in eukaryotes.

## 2. Nuclear Membrane Proteins and Heterochromatin Formation in Fission Yeast

### 2.1. Organization of Chromosome in the Nucleus

*S. pombe* cells grow with a haploid genome consisting of three chromosomes: I, II, and III [36]. These chromosomes have a centromere at their middle and telomere repeat sequences at both ends; rDNA repeats, which code for ribosomal RNA, are present at both ends of chromosome III flanked by telomere repeat sequences (Figure 1A). In mitotic interphase, these chromosomes are packed in the hemispherical half of the nucleus; the other half of the nucleus is rich in RNA, corresponding to the nucleolus in higher eukaryotes; two protrusions of chromatin are embedded in the RNA-rich hemisphere (Figure 1B). Centromeres are associated with the spindle pole body (SPB; a centrosome-equivalent structure in fungi) located on the cytoplasmic side of NE, and telomeres are on the NE near the nucleolus located at the opposite side of the SPB [37,38,39,40,41,42,43] (Figure 1C).

### 2.2. Heterochromatin as Transcriptionally Silent Regions

In metazoan cells, a typical electron-dense heterochromatin is observed beneath the NE or around the nucleoli [44]. Such an electron-dense heterochromatin is hardly seen in the nucleus of *S. pombe*. In this organism, heterochromatic regions have been identified as transcriptionally silent chromatin regions: The pericentromeric region, the sub-telomeric region, and the silent mating type locus (hereafter referred to as *mat* locus) [45,46,47]. These heterochromatic regions in *S. pombe* share histone modification marks (post-translational modifications such as methylation and acetylation) similar to those in other eukaryotes, but do not completely match with the cytological definition of heterochromatin as condensed chromatin domains [48]. These regions are located proximally to the NE (Figure 1C), and the NE proteins anchoring them to the NE are described in Section 4.

### 2.3. “Knob” Regions

A distinct region of chromatin was recognized as a “knob,” which shows a highly condensed chromatin region near the sub-telomeric region in *S. pombe* [48] (Figure 1A,C; also see Figure 4). The “knob” region does not share typical histone modification marks of heterochromatin, but matches the cytological definition of heterochromatin and shows unique properties different from other constitutive heterochromatin regions [39]. In this review, we define the knob region as heterochromatin and introduce its properties in a later section (see Section 3.4). 

## 3. Mechanisms for Heterochromatin Formation in Fission Yeast

In *S. pombe*, *mat* locus was first recognized as the most striking region of gene silencing [49] and was predicted as an inheritable element for silencing [50], which is now recognized as an epigenetic regulation. A group of proteins required for the inheritance of mating types was identified as Swi1‒Swi9 [49]. Among them, Swi6 was identified as a key player in the epigenetic regulation of mating types [51] and was later identified as a homolog of the heterochromatin protein 1 (HP1) in *Drosophila* and mammals [52]. These pioneering studies have contributed to the unveiling of the molecular mechanisms underlying heterochromatin formation [45,46,47,53,54]. In this section, we summarize the molecular machineries involved in heterochromatin formation. 

### 3.1. RNAi-Mediated Silencing Machinery

In *S. pombe*, constitutive heterochromatin is nucleated, propagated, and maintained at limited genomic loci such as the pericentromere, the sub-telomere, and the *mat* locus, all of which contain a repetitive DNA sequence. To accomplish heterochromatin formation in these regions, RNA processing and chromatin modification play a pivotal role in a mutually dependent manner (Figure 2). RNA interference (RNAi) is a post-transcriptional gene silencing (PTGS) regulation that has been initially considered as a system for degrading protein-coding transcripts [55,56]. However, genetic analyses in *S. pombe* led to the intriguing finding that the core components of RNAi machinery are essential for pericentromeric heterochromatin formation [57]. With this discovery, numerous studies have revealed the function of RNAi and its interplay with chromatin proteins in the context of heterochromatin formation [47,58].

PTGS involves long non-coding RNAs (lncRNAs) transcribed by RNA polymerase II (Pol II) from repetitive sequences located in the heterochromatic regions: *dg*/*dh*, *dh*-like, and *cenH* for pericentromere, sub-telomere, and *mat* locus, respectively [57,59,60,61]. lncRNAs are cleaved by Dicer (Dcr1), generating small interfering RNA (siRNA). siRNAs are incorporated into an RNA-induced initiation of transcriptional silencing (RITS) complex, which consists of Ago1, Chp1, and Tas3 [62]. Ago1 belongs to the Argonaute family of proteins that can directly interact with siRNAs. RITS complex recruits an RNA-directed RNA polymerase complex (RDRC) that facilitates siRNA generation with Dicer to increase siRNA-bound RITS complex [63,64]. Thus, RITS complex functions as a guide for nucleation of heterochromatin assembly. RITS complex is also important in recruiting a Clr4 methyltransferase complex (CLRC) consisting of Clr4, Rik1, Cul4, Raf1, and Raf2, through an interaction with Stc1 [64,65]. Clr4 is the sole methyltransferase for the 9th lysine residue of histone H3 (H3K9) in *S. pombe* and shares functional similarity with mammalian Suv39h [66,67]. Note that H3K9 dimethylation (H3K9me2), instead of trimethylation (H3K9me3), is a major heterochromatic mark in *S. pombe*. H3K9me2 assembles chromodomain-containing proteins, such as Chp1, Chp2, and Swi6 (homologs of human HP1) [68]. Once histones are marked by H3K9me2, a positive feedback loop reinforces heterochromatin assembly: siRNA-bound RITS complex stably binds to H3K9me2 via the chromodomain of Chp1, and consequently accelerates CLRC and Swi6 recruitment [62]. Conversely, it has been shown that siRNA generation and localization of RNAi components to heterochromatin is dependent on chromatin factors, including Clr4 and HP1 homologs [69]. Therefore, RNAi and chromatin factors function in an interdependent manner with regards to heterochromatin formation, thereby raising a “Chicken and Egg” problem of which one works first [70]. The finding that Ago1 can bind with heterochromatin- and Dicer-independent species of *dg* siRNAs provides a clue to solving this conundrum, suggesting that the primal amplification of siRNAs by RNAi machinery acts as a seed to create heterochromatin, even in the absence of heterochromatin. Dcr1 is localized to NE through interaction with NPC components [34] (Figure 2; see Section 6.1).

In addition to the RNAi machinery, RNAi-independent mechanisms involving nuclear RNA processing and degradation factors such as the TRAMP (Trf4-Air1/Air2-Mtr4 polyadenylation) complex also contribute to establishing heterochromatin. TRAMP containing Cid14, a member of the Trf4 family of poly(A) polymerases, targets RNAs into degradation machineries that include the exosome. Both RNAi-dependent and RNAi-independent mechanisms work in parallel to target CLRC to the repetitive DNA sequences located in the constitutive heterochromatin domains [71,72,73,74,75,76,77].

### 3.2. HDAC-Mediated Silencing Machinery

Transcriptional gene silencing (TGS) pathways also contribute to heterochromatin formation in addition to the above RNAi-mediated PTGS machinery (Figure 2). Chp2 and Swi6 bound to H3K9me2 can target SHREC, a class II histone deacetylase (HDAC) complex containing Clr3 and Mit1 (Snf2 family ATPase) [78]. Swi6 can also recruit another protein complex containing class I HDAC Clr6 to the heterochromatin region [79]. Hypoacetylation of the lysine residues of histone H3 and H4 via HDAC, which is localized at the heterochromatin region, suppresses chromatin remodeling, restricting the access of Pol II to suppress transcription. Recent studies have reported that Swi6/HP1 protein contributes to the liquid-liquid phase separation associated with the reshaping of the nucleosome core, which accordingly quarantines heterochromatin to restrict Pol II accessibility [80]. 

In addition to these general TGS pathways, it is also known in *mat* locus that several DNA binding proteins, such as CENP-B homologs (Abp1 and Cbh1) and ATF/CREB family proteins (Atf1 and Pcr1), recruit HDACs to their binding sites within the locus [81,82,83]. 

Ubiquitination of histone H3 also affects heterochromatin formation through the modulation of H3K9 methylation. Cul4, Rik1, and Raf1 of *S. pombe* CLRC are presumed to have a ubiquitin E3 ligase activity owing to their structural similarity with CUL4-DDB1-DDB2 of other eukaryotes [84,85]. Indeed, a recent report demonstrates that CLRC ubiquitylates lysine 14 of histone H3 (H3K14ub) and H3K14ub promotes H3K9 methylation by Clr4 [86]. 

### 3.3. Boundary Elements Between Heterochromatin and Euchromatin

The propagation of the heterochromatin is strongly restricted within its locus to avoid the leakage of the silencing effect on the neighboring genes. Inverted repeat sequences called *IR*s (*IRC* or *IR-L/R* at the pericentromere or *mat* locus, respectively) work as boundaries in demarcating the heterochromatin and euchromatin regions [59,87,88] (Figure 3). The function of this boundary is partly conferred by the action of TFIIIC, an RNA polymerase III (Pol III) transcription initiation factor, but in a Pol III transcription-independent manner [89]. DNA regions including *IR*s in which TFIIIC binds irrelevant to Pol III tend to form a cluster and accumulate around the nuclear periphery [89]. This peripheral nuclear tethering is supposed to be necessary in its function as a boundary.

The border between the heterochromatin and euchromatin is also controlled by the anti-silencing factor, Epe1 [90,91]. Epe1 has a JmjC-domain, which is the catalytic domain of histone demethylase; however, demethylase activity in vitro has not been detected in *S. pombe* Epe1 [91,92,93]. Epe1 is recruited to the heterochromatic regions in a Swi6-dependent manner [94] but enriched only around the boundary elements owing to the degradation by the proteasome inside the heterochromatin region [95] (Figure 3). It has been proposed that Epe1 might modulate H3K9 methylation indirectly, thus, compelling the localization of HDAC Clr3 bound to Swi6 [96,97,98]. Epe1 also recruits SAGA, a histone acetyltransferase complex, to counteract HDAC Sir2 [99]. These combined activities prevent the spreading of heterochromatin beyond the boundary (Figure 3). TFIIIC and Epe1 have been suggested to act redundantly for effective boundary formation [100].

### 3.4. DAPI-Dense “Knob” Region

A “knob” is recognized as a distinct region of highly condensed chromatin that is densely stained with a DNA-specific fluorescent dye, 4′,6-diamidino-2-phenylindole (DAPI). This highly condensed chromatin is formed near the telomere, spanning approximately 50 kb next to the sub-telomeric heterochromatin regions of chromosomes I and II [48], which corresponds to a special chromatin region called “ST-chromatin” [101]. This knob region (ST-chromatin) shows characteristic histone modification patterns: Low levels of methylation at H3K4, H3K9, and H3K36 and acetylation at H3K14, H4K5, H4K12, and H4K16 [48,101,102] (Figure 4). Despite the low levels of H3K36 methylation, interestingly, knob formation requires Set2, a sole methyltransferase of H3K36 in *S. pombe*. It has also been shown that the kinetochore protein Sgo2, FACT (facilitates chromatin transcription), which is an essential histone chaperon consisting of Spt16 and Pob3, and the mono ubiquitinated histone H2B (H2Bub) are required for knob formation [102,103]. H2Bub facilitates histone H2A/H2B dimer deposition by FACT. In addition, FACT maintains the nucleosome density in the sub-telomeric region [103], which may be the key to knob formation.

Genes inside the knob region are transcribed at a very low level in vegetative cells, but they can be upregulated upon nitrogen starvation, accompanied with a loss of knob formation [48,104]. Curiously, a reporter gene inserted within the knob region of the right arm of chromosome II, but not in the right arm of chromosome I, was slightly silenced, depending on Sgo2 and Set2 [48,102]. These results suggest that the knob region has the potential to locally suppress gene expression, depending on the adequate localization pattern of Sgo2 and histone modifications. Deletion of Set2 or Sgo2 not only increases the expression of the genes located in the knob but also alters the replication timing of knob chromatin from late to early S-phase [102], suggesting that knob condensation appears to be important in modulating transcription and DNA replication. 

Intriguingly, neocentromere, which is a newly generated centromere that escaped the catastrophe of an authentic centromere deletion, is frequently formed in this knob region [105]. Neocentromere formation requires RNAi-machinery-mediated heterochromatin formation in sub-telomeric region [105]. Therefore, knob is a unique chromatin structure that has the potential to provide a flexible platform in response to various cellular reactions.

## 4. Proteins Attaching Heterochromatic Regions to the NE

INM proteins play a key role in attaching heterochromatin loci and facilitating/maintaining heterochromatin formation [106]. Heterochromatic regions in *S. pombe* are located proximally to NE, as shown in Figure 1C. In this section, we describe *S. pombe* NE proteins that attach heterochromatic regions to the NE.

### 4.1. Proteins Attaching Heterochromatic Regions to the NE in Mitosis

Centromeres are attached to the SPB through the interaction with Sad1/Unc84 (SUN)-domain protein Sad1 [107]. Sad1 interacts with the outer kinetochore through the interaction with Csi1 [108,109]. Sad1 also interacts with Klarsicht/ANC-1/Syne homology (KASH) domain-containing proteins Kms1/2 and forms a complex termed as “linker of nucleoskeleton and cytoskeleton (LINC) complex,” which is localized exclusively at the SPB during mitotic interphase [110,111,112,113]. Centromeres detach from the SPB in Lem2 and Csi1 double deletion mutants [33,114]. Thus, centromeres are attached to the SPB through interactions among Lem2, Csi1, and the LINC complex across the NE (Figure 5A).

Telomeres are connected to the NE through the interaction of the INM protein, Bqt4, with the telomere protein, Rap1 [115] (Figure 5B). Mutation or loss of these proteins causes dissociation of telomeres from the NE [109,115]. Telomeres also detach from the NE in the absence of Lem2 [33,116], and the double deletion of Lem2 and Dsh1 exacerbates the phenotype of telomere detachment, suggesting that Lem2 and Dsh1 play a role in attaching telomeres to the NE [33].

The chromatin region containing the *mat* locus is localized at the nuclear periphery through the cluster formation of boundary elements on the inverted repeat sequence *IR-L/R* [89]. A recent report demonstrated that the NE localization of *mat* locus is mediated through an interaction with nucleoporin Amo1 [31] (see Section 6.2).

Knob is also located near the NE, but in a Bqt4-independent manner [117]. A conserved subfamily of SNF2 chromatin remodeler protein Fft3 is involved in the localization of knob at the NE because loss of Fft3 results in the detachment of the sub-telomeric region from the NE [117]. Fft3, which binds to a solo long-terminal repeat (LTR) located at the border region between knob and euchromatin, interacts with a LEM/HEH-domain inner nuclear membrane protein Man1 [117], suggesting that an interaction between Fft3 and Man1 is involved in attaching knob to the NE.

### 4.2. Proteins Attaching Heterochromatic Regions to the NE in Meiosis

As described in Section 2.1 (Figure 1), centromeres form a cluster with the SPB, while telomeres are located on the opposite side of the nucleus in mitotic interphase (the configuration called a Rabl orientation). Upon entering meiosis, centromeres and telomeres switch their positions in the nucleus: Telomeres become associated with the SPB, while centromeres detach from the SPB (called a bouquet configuration). During the process of centromere–telomere switching, meiosis-specific telomere proteins, Bqt1 and Bqt2, interact with the telomere protein Rap1. Bqt1 and Bqt2 also interact with the NE protein complex, Sad1, and Kms1 (LINC complex), to attach telomeres to the NE [118]. Kms1 interacts with cytoplasmic dynein motor complex on microtubules, which tethers telomeres toward the SPB across the NE [119,120] (Figure 5C). This phenomenon provides a striking example, in which interactions between chromatin and NE play an important role in the organization of the chromosomes within the nucleus.

## 5. NE Proteins Modulating Heterochromatin Formation

### 5.1. Lem2 Functions in Heterochromatin Formation

#### 5.1.1. Lem2 is a Conserved Protein

Lem2 is one of the most broadly conserved INM proteins, sharing two transmembrane (TM) domains and a MAN1/Src1 C-terminal (MSC) domain at the C-terminal region [3,11,121] (Figure 6A). By in silico homology search using MSC domain as a query followed by experimental localization analysis, members of the Lem2 protein family were found in different eukaryotic supergroups: Opisthokonta (yeasts to humans) [11], Amoebozoa (Src1 in *Dictyostelium discoideum*) [122,123], SAR (Lem2 and MicLem2 in *Tetrahymena thermophila*) [13], and Archaeplastida (plants) [13]. Lem2 in metazoans has the LEM domain and Lem2 in fungus has the LEM-like HEH (helix-extended-helix) domain at the N-terminal region; no obvious LEM or HEH domains were found in other supergroups.

Since overexpression or deletion of Lem2 in *S. pombe* alters chromosome functions and NE morphology and integrity, Lem2 plays an important role in both chromosome and NE functions [33,35,107]. Recent reports show that Lem2 interacts with chromatin and regulates its functions, especially in heterochromatin formation [33,35,124]. This function seems to be unique in Lem2 because the deletion of Man1 or Ima1 does not show any effect on heterochromatin formation [33].

#### 5.1.2. Lem2 on Centromeric Heterochromatin

One of the most prominent phenotypes of the Lem2 deletion mutant is severe chromosome instability in a nutrient-dependent manner [35]. Deletion of Lem2, but not Man1 and Ima1, results in a high rate of minichromosome loss; the phenotype appears only in rich medium culture conditions. In contrast, the deletion of Csi1, which is a crucial protein that attaches centromeres to the SPB, displays a high rate of minichromosome loss in both rich and minimum medium culture conditions. This high loss rate is probably caused by the deficiency of pericentromeric heterochromatin formation. Although a genome-wide mapping for the Lem2-binding region has not been made public thus far, some Lem2-binding regions have been reported, and one of the major binding sites of Lem2 was the central core in the centromere [33,124]. Centromeres in *S. pombe* are composed of a central domain flanked with outer repeats. The central domain is divided into a central core (*cnt*) and an innermost repeat (*imr*), where CENP-A (centromere-specific histone H3 variant)-containing nucleosome and kinetochore are formed [125,126]. The outer repeat consists of inversed repeat elements named *dg* and *dh* [127]. Pericentromeric heterochromatin is generated in this *dg*/*dh* region in an RNAi and TGS-dependent manner (see Section 3.1). Lem2 binds to a central core through its N-terminal region, including HEH domain, and this localization depends on the nutrient condition (Figure 6B): Lem2 localizes to this region only in rich medium culture condition [33,35,116,124,128]. The deletion of Clr4 impairs Lem2 localization at the central core, implying that heterochromatin formation is required for proper Lem2 localization [124]. Interestingly, localizing Lem2 to the central core is required for heterochromatin augmentation at the pericentromere [35], indicating that Lem2 can function as a positive feedback loop to promote heterochromatin formation. Loss of Lem2 causes a decrease in the pericentromeric H3K9me2 and a de-repression of pericentric silencing, particularly when combined with the deletion of Dsh1, suggesting that Lem2 contributes to heterochromatin formation independently of Dsh1-mediated RNAi pathway [33]. The Lem2-binding INM protein Nur1 displays similar effects in this region [33,124].

#### 5.1.3. Lem2 on Telomeric Heterochromatin

Lem2 also interacts with chromatin regions close to the sub-telomeric heterochromatin region [124]. ChIP-seq and ChIP-chip analyses show that Lem2 binds to the telomere side of sub-telomeric heterochromatin at *tel3L* and *tel3R*, but no significant enrichment at *tel1L* and *tel2L* regions [33,124,129]. Loss of Lem2 shows de-repression of *tlh1/2* genes located within the sub-telomeric region but does not affect H3K9me2 levels in this region [33]. Furthermore, Lem2 regulates the balance of SHREC and Epe1 binding at heterochromatic regions [33,124]. In the absence of Lem2, the SHREC component dissociates from the heterochromatic gene silencing sites while Epe1 associates with the sites. Up-regulation of *tlh1/2* genes by Lem2 deletion is bypassed by Epe1 deletion, indicating that gene silencing by Lem2 in sub-telomeric region largely depends on SHREC-Epe1-mediated pathway without affecting H3K9me2 [33].

#### 5.1.4. Lem2 on LTR Sequences

Deletion of Lem2 is likely to increase the recombination between LTRs [33,35]. LTRs found at the end of retrotransposons are used when a virus integrates its genomic DNA into its host genome [130]. Expression of retrotransposable elements must be strongly regulated to prevent undesired pop-out and pop-in of the retrotransposons causing genome instability [117,131,132]. In the Lem2 deletion mutant, the expression of LTRs at the sub-telomeric region and frequency of recombination at LTRs are increased [33,35], suggesting that Lem2 plays a role in suppressing the expression of LTRs.

#### 5.1.5. Molecular Domains of Lem2 for Heterochromatin Functions

C-terminal region of Lem2, containing two transmembrane domains and MSC domain, is sufficient for heterochromatic gene silencing at both pericentromeric and sub-telomeric regions [33,35]. An MSC domain without the transmembrane domain cannot suppress gene expression, indicating that membrane localization is essential for gene silencing activity of Lem2 [33]. In contrast, the N-terminal region of Lem2 does not show gene silencing activity. This domain restores the centromere association with the SPB in the absence of Lem2 and Csi1 [114], whereas it does not restore telomere association in the absence of Lem2 and Dsh1 [33]. Thus, the N-terminal region functions to anchor only the centromeric region to the NE.

### 5.2. Regulation of Lem2 Localization

Lem2 localizes at the INM and beneath the SPB, but biased to the SPB [128,129]. These localizations depend on Csi1 for the SPB and Bqt4 for the NE, respectively. The Deletion of Csi1 causes the disappearance of Lem2 from the SPB, although the direct interaction between Lem2 and Csi1 has not been reported. In contrast, the deletion of Bqt4 induces strong SPB accumulation of Lem2, indicating that Bqt4 plays a role in retaining Lem2 at the NE [128,129]. Lem2 and Bqt4 bind directly through an interaction between Bqt4-binding motif ((D/E)_3-4_xFxxxɸ) in Lem2, located at the nucleoplasmic region near the first transmembrane domain and APSES domain in Bqt4 [128,133,134]. This domain binds to DNA and also interacts with the Bqt4-binding motif shared in Lem2, Sad1, and Rap1 in a competitive manner [133]. In cells with deleted Csi1 and Bqt4, Lem2 shows a biased localization to the SPB, suggesting that other factors are also involved in the Lem2 localization to the SPB [129]. One of those factors appears to be Nur1 because Lem2 and Nur1 depend on each other for SPB localization in fission yeast *Schizosaccharomyces japonicus* [135]. During mitosis, Lem2 disappears from the SPB at prophase and reappears to the SPB at the onset of anaphase [136]. Conversely, Ima1 appears on the SPB at prophase and disappears at the onset of anaphase [136], implying that Lem2 and Ima1 have cell cycle-specific functions on the SPB during mitosis.

### 5.3. Membrane Protein Network Regulating Lem2 Functions

#### 5.3.1. Lem2 Functions through Lnp1

The deletion of Lem2 causes centromeric defects such as defective formation of pericentromeric heterochromatin and chromosome instability, all of which are suppressed by the duplication of the *lnp1* gene [35]. Lnp1 (homolog of human Lunapark) is an ER membrane protein conserved from yeasts to humans. Lnp1 localizes at the three-way junction of tubular ER network in humans and *S. cerevisiae* [137,138,139,140,141]. Double deletion of Lem2 and Lnp1 disturbs the partitioning between NE and ER, and consequently causes severe membrane disorder and growth defects [142]. Moreover, Lem2 and Lnp1 act as barriers to the membrane flow between the ER and Golgi and contribute to the control of the nuclear size [143]. Thus, Lnp1 is likely involved in maintaining chromosome integrity through an indirect pathway, in which Lnp1 and Lem2 cooperatively maintain the NE/ER boundary and NE integrity.

#### 5.3.2. Lem2 Functions through Bqt4

Loss of Lem2 and Bqt4 confers synthetic lethality [35]. In this double mutant, leakage of nuclear proteins frequently occurs owing to NE breakage, leading to cell death [144]. The domains responsible for this genetic interaction are the MSC domain of Lem2, and the N-terminal and transmembrane domains of Bqt4. In addition, the transmembrane domains of both proteins are essential for their effect, suggesting that both proteins are required for their functions on the NE. The synthetic lethal phenotype of the Lem2 Bqt4 double deletion is suppressed by the very-long-chain fatty acid elongase Elo2, which is essential for *S. pombe* cell viability [144]. Elo2 synthesizes “very-long-chain” fatty acids (chain length of carbon atoms longer than 21), which play crucial roles that cannot be substituted by “long-chain” fatty acids (chain length of carbon atoms 11-20). Most of the very-long-chain fatty acids constitute sphingolipids, which play important roles in maintaining the skin barrier and myelin sheath in mammals [145,146,147]. The overexpression of Elo2 suppresses the defective formation of the pericentromeric heterochromatin and chromosome instability caused by the loss of Lem2, but does not restore the telomeric attachment to the NE by the loss of Bqt4, indicating that Elo2 bypasses Lem2 functions. Moreover, the loss of *S. pombe* Elo2 is complemented by an overexpression of human orthologs, suggesting conserved roles of Elo2 in genome stability [144].

#### 5.3.3. Lem2 Functions through the ESCRT-III Complex

Loss of Lem2, in combination with loss of Bqt4 or Lnp1, causes severe nuclear protein leakage owing to NE holes formed [142,144]. Recent studies indicate that Lem2 seals an NE hole in cooperation with Cmp7 (homologue of human CHMP7) and endosomal sorting complex required for transport-III (ESCRT-III) [148,149,150]. Lem2 may play a role in sealing the NE by liquid phase separation [151]. Unrepaired NE holes can cause DNA damage beneath the hole [150], and aberrant accumulation of ESCRT-III in the NE hole is observed with a DNA damage marker protein in micronuclei of human cells [152], suggesting that sealing the NE holes is the critical process to maintain chromosomes. Considering that Lem2 has a role in facilitating checkpoint signaling in response to replication stress [153], Lem2 may function in DNA damage repair at the holes. On the other hand, Vps4, which is an AAA-ATPase, disassembles the ESCRT-III complex and releases the pericentromeric heterochromatin from Lem2 at the end of mitosis in *S. japonicus*, indicating that Lem2 and ESCRT-III complex function in remodeling the attachment of heterochromatic regions to the NE [135]. Lem2 possibly ensures genome stability through maintaining the NE integrity in fission yeasts.

## 6. Nucleoporins and Heterochromatin

The NPC is composed of several subcomplexes, namely, the outer ring, inner ring, central channel, nuclear basket, and the cytoplasmic filament (Figure 7). These subcomplexes associate with the nuclear membrane by interacting with the transmembrane Nups to form cylindrical structures connecting the nucleoplasm and cytoplasm [19,22,23,24,25,26,27,28,29,154,155]. This typical organization of NPC is conserved among eukaryotes, although it exhibits species-specific variations in *S. pombe* [23,156]. Nups also contribute to chromatin organization in the nucleus. In *S. pombe*, some of the Nups are reported to be involved in gene silencing [31,32].

### 6.1. Nucleoporins Modulate Gene Silencing

Dcr1, a key regulator of RNAi-mediated gene silencing, has been shown to be localized at the NPC; importantly, this localization is essential for pericentromeric heterochromatin formation [34].

Moreover, Dcr1 and Nup85 (one of the outer ring Nups) are enriched in the promoter regions of stress response genes [157]. This implies that the transcription of stress response genes at the NPC is repressed by Dcr1; however, the relevance of NPC localization of Dcr1 to a functional RNAi-dependent gene silencing pathway remains elusive. In addition, the interaction of a CLRC complex component Raf1/Clr8 with a gene product of *nup189^+^* was observed in yeast two-hybrid assay [158]. Proteome analysis of immunoprecipitated Swi6 fraction also identified several Nups, including inner ring Nups, Npp106, and Nup186 [32]. Npp106 and Nup211 (a nuclear basket Nup) accumulated at the pericentromeric regions [32]; Npp106 mutant showed a defect in gene silencing at pericentromeric heterochromatin [32]. It has also been reported that most of the tRNA loci, including the boundary element of pericentromeric heterochromatin, associate with Nup85 [117]. Moreover, the functional role of *Drosophila* Nup93, an inner ring Nups, in the silencing of Polycomb target genes was recently reported [159]. These cumulative pieces of evidence strongly suggest the involvement of NPC in the regulation of gene silencing in heterochromatic regions.

### 6.2. Nucleoporin Amo1 Sequesters Heterochromatin to the NE

Amo1 has been identified as a factor affecting cytoplasmic microtubule organization [160] and is localized at the cytoplasmic side of NPC [156]. Interestingly, Amo1 was recently reported to participate in heterochromatin formation at *mat* locus and pericentromeric regions [31]. Amo1 can bind to Rix1, a component of an RNA processing complex (RIX complex), which localizes to heterochromatin regions. Through interactions with Rix1, Amo1 can recruit the *mat* locus to the nuclear periphery. Importantly, this interaction is required for peripheral localization and silencing of ectopically induced heterochromatin. Amo1 also interacts with the FACT histone chaperon complex that binds to Swi6 and is required for heterochromatin formation [161,162]. Therefore, Amo1 is thought to facilitate the efficient loading of FACT onto Swi6-bound heterochromatin to suppress histone turnover and maintain the epigenetic state of the heterochromatin [31]. It is also suggested that Amo1, which localizes at a nuclear periphery different from NPC, may function in the regulation of heterochromatin maintenance [31].

## 7. Perspectives

In this review, we have described NE proteins that are involved in heterochromatin formation and functions in *S. pombe*. An important role for the NE is to provide a physicochemical platform for chromosomes, and tethering chromatin to their vicinity evokes chromatin potential to modulate its genetic activities. For example, FG repeat-containing proteins in NPCs produce an amphiphilic field by forming hydrogel or liquid droplets [163]. Thus, an intriguing possibility is that FG repeat proteins and Swi6, which also form liquid droplets [80], create a peripheral compartment through phase separation, which is required for the silencing of heterochromatin. Recently, other interesting studies have demonstrated that lipid metabolism is involved in modulating heterochromatin states and chromatin functions [142,164]. The involvement of the lipid components of NE and ER in chromatin functions need to be elucidated. Considering that the very-long-chain fatty acid elongase Elo2 could compensate for the loss of Lem2 and its interaction with NE proteins, one could speculate that a specific lipid generated by a membrane protein network focusing on Lem2 could have a yet unknown role in heterochromatin formation and functions.

## Figures and Tables

**Figure 1 cells-09-01908-f001:**
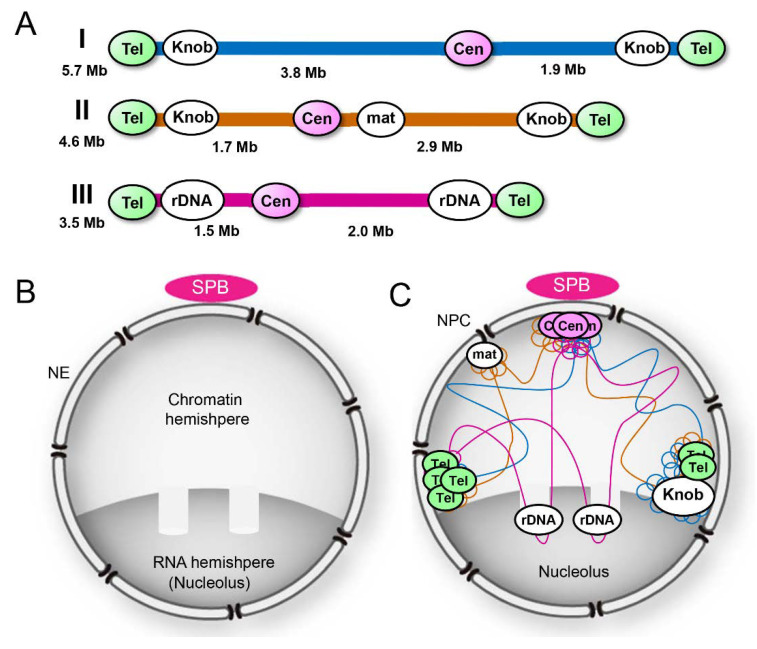
The overall structure of the *Schizosaccharomyces pombe* nucleus. (**A**) Genomic organization of *S. pombe* chromosomes. Cen, centromere; Tel, telomere; mat, *mat* locus; and rDNA, repeat sequences coding ribosomal RNA. (**B**) *S. pombe* nucleus: A chromatin hemisphere and an RNA hemisphere with two protrusions of chromatin. (**C**) Spatial organization of chromosomes in the nucleus.

**Figure 2 cells-09-01908-f002:**
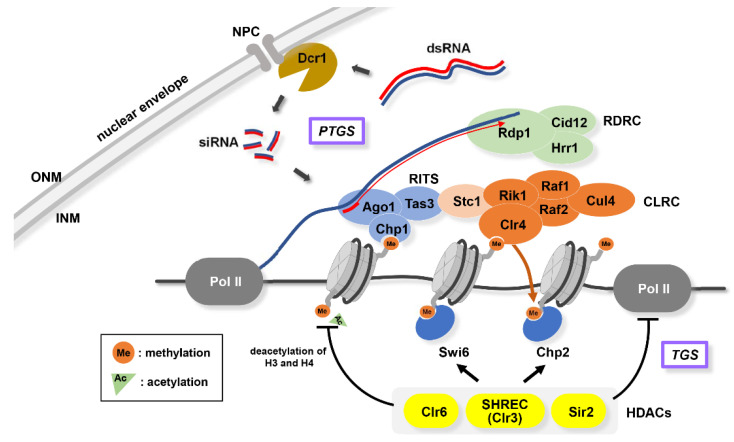
Principles of heterochromatin formation. RNAi-mediated PTGS (post-transcriptional gene silencing) and HDAC-mediated TGS (transcriptional gene silencing). PTGS involves Dicer (Dcr1) located at the NPC; RITS (RNA-induced initiation of transcriptional silencing) complex consist of Ago1, Chp1, and Tas3; RDRC (RNA-dependent RNA polymerase complex); CLRC (Clr4 methyltransferase complex) consist of Clr4, Rik1, Cul4, Raf1, and Raf2. TGS involves HDACs (Clr6, SHREC, and Sir2) and HP1 (Swi6 and Chp2).

**Figure 3 cells-09-01908-f003:**
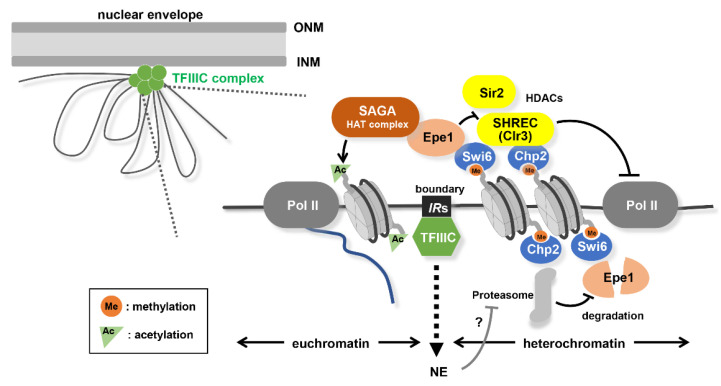
Formation of heterochromatin/euchromatin boundary. Boundary elements (*IR*s) clustered at the nuclear periphery through TFIIIC binding. Epe1 (anti-silencing factor) is specifically accumulated at the boundary (*IR*s) owing to the proteasome-mediated selective elimination of Epe1 from heterochromatin. A heterochromatin/euchromatin boundary is formed owing to a balance between the opposing activity of histone acetyltransferase (SAGA recruited by Epe1) and histone deacetylase (SHREC or Sir2 recruited by Swi6 and Chp2).

**Figure 4 cells-09-01908-f004:**
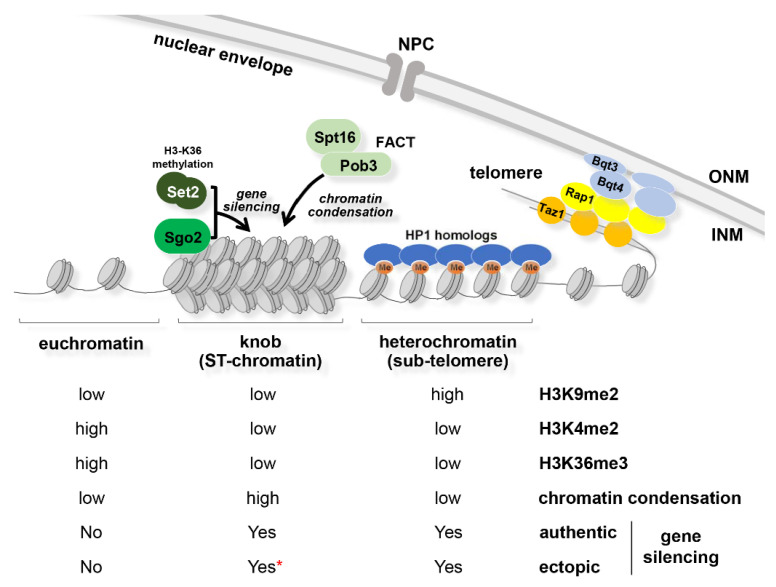
Chromatin states in telomeric regions. A highly condensed “knob” at ST-chromatin region formed by a cooperative action of Sgo2 and Set2, and a FACT-dependent nucleosome assembly pathway. (Lower columns) Levels of histone modifications (H3K9me2, H3K4me2, and H3K36me3). Levels of chromatin condensation. Gene silencing: “authentic”, expression of endogenous genes upon nitrogen starvation; “ectopic”, expression of the *ura4*^+^ reporter gene ectopically inserted; * silencing in the knob region depends on the position of *ura4*^+^ insertion.

**Figure 5 cells-09-01908-f005:**
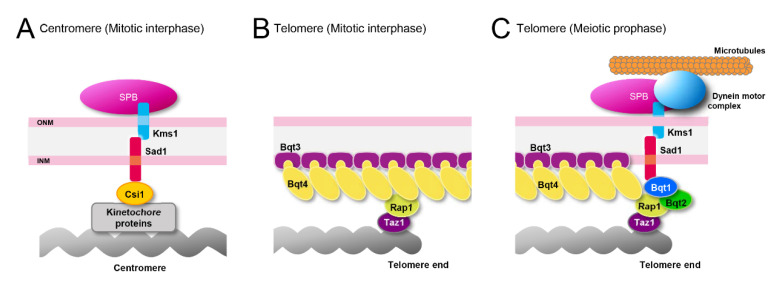
NE proteins attaching heterochromatin regions to the NE. (**A**) Centromeres attached to the SPB in mitotic interphase. (**B**) Telomeres attached to the NE in mitotic interphase. (**C**) Telomeres attached to the SPB in meiotic prophase.

**Figure 6 cells-09-01908-f006:**
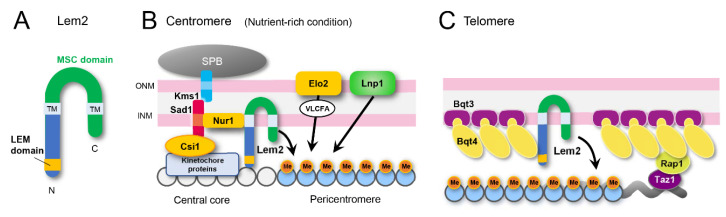
Lem2 for heterochromatin formation. (**A**) Molecular domains of Lem2. TM, transmembrane domain. (**B**) Lem2 on pericentromeric heterochromatin. White circles represent chromatin at the central core of the centromere and blue circles represent chromatin at the pericentromeric heterochromatin. Lem2 is localized at the central core and promote heterochromatin formation at the pericentromeric regions. Elo2 and Lnp1 compensate Lem2 functions. VLCFA, very-long-chain fatty acid. (**C**) Lem2 on sub-telomeric heterochromatin. Lem2 is localized to the NE through interaction with Bqt4, and promotes heterochromatin formation at the sub-telomeric regions attached to the NE through an interaction between Rap1 and Bqt4.

**Figure 7 cells-09-01908-f007:**
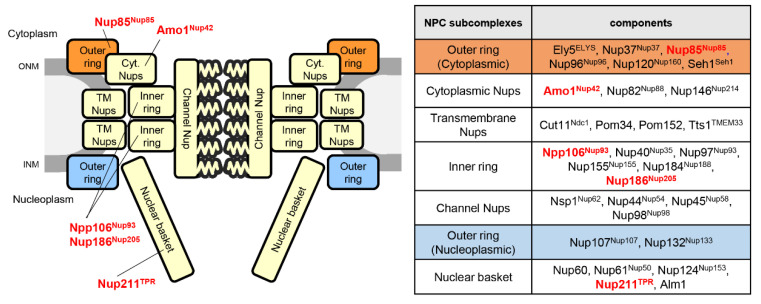
Subcomplex structure of the *S. pombe* NPC. (**Left**) Schematic drawing of the NPC. Nups described in the text are shown in red; human orthologs are written in superscript. TM Nup, transmembrane Nup; ONM, outer nuclear membrane; INM, inner nuclear membrane. (**Right**) NPC subcomplexes and their components.

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
