# Peer review of "Nuclear Envelope Proteins Modulating the Heterochromatin Formation and Functions in Fission Yeast"

_cells, 2020, doi:10.3390/cells9081908_

Round 1

Reviewer 1 Report

This review manuscript by Drs Hiraoka, Haraguchi and their colleagues is a well-written article. Although focusing on fission yeast, it covers the field quite comprehensively. The review should be therefore very informative to the general readership of Cells. There are only a few minor errors or suggestions that are listed below.

  • P1, L37. Delete “it”.
  • P2, L49. It would be better to mention the proteins in “in addition to the three proteins described above”.
  • P5, L180-181. Please pay attention to the formatting issues.
  • Section 5.3.3. Lem2 may seal the NE by liquid phase separation (PMID: 32494070).
  • P11, L421-422 extra spaces at the beginning.

Author Response

(Comment 1) P1, L37. Delete “it”.

(Response 1) Thank you for pointing this out. We deleted “it” (line 37):

“The nuclear lamina exists beneath the INM, and is known to play an important role ---.”

(Comment 2) P2, L49. It would be better to mention the proteins in “in addition to the three proteins described above”.

(Response 2) Thank you for the suggestion.We mentioned the name of proteins (line 49):

“in addition to the three proteins (Lap2, emerin and MAN1) described above.

(Comment 3) P5, L180-181. Please pay attention to the formatting issues.

(Response 3) Thank you. We formatted.

(Comment 4) Section 5.3.3. Lem2 may seal the NE by liquid phase separation (PMID: 32494070).

(Response 4) Thank you for the suggestion. We added the statement with the citation as suggested (lines 410-411):

Lem2 may play a role in sealing the NE by liquid phase separation [151].

(Comment 5) P11, L421-422 extra spaces at the beginning.

(Response 5) Thank you. We formatted.

Reviewer 2 Report

Excellent and comprehensive review that merits publication. The manuscript is well-written and includes nice figures. I only find minor issues that needs to be addressed prior publication as follows:

_An accumulation of evidence has shown that heterochromatin assembly is transcription-dependent as conserved RNA processing machinery, including RNAi –dependent and RNAi-independent mechanisms, targets transcripts such as long non-coding RNA to direct assembly of both constitutive and facultative heterochromatin. However, I do not notice any mention about RNAi-independent pathways. Please mention such mechanisms and cite some of the relevant recent literature, e.g. Marina et al. 2013 (PMID: 24013500); Touat-Todeschini et al. 2017 (PMID: 28765164); Vo et al. 2019 (PMID: 31269446); Lee et al. 2020 (PMID:32101745).

_ Lines 120-132: When describe RNAi machinery please acknowledge recent high-profile papers by Roche and Martienssen 2016 (PMID: 27738016) and Folco et al. 2017 (PMID: 28199302) that underscore the role of RNAi in cell differentiation and chromosome segregation, respectively.

_ Fig. 7: It will be useful to non-Pombe readers to have the name of the human orthologs in superscript (e.g. Amo1NUP42; Npp106NUP93).

_ Line 466: Please state “other interesting findings…” instead of “another…”. Also, the sentence is a bit awkward and may be written as “Recently, other interesting studies have demonstrated that...”

Author Response

(Comment 1) An accumulation of evidence has shown that heterochromatin assembly is transcription-dependent as conserved RNA processing machinery, including RNAi –dependent and RNAi-independent mechanisms, targets transcripts such as long non-coding RNA to direct assembly of both constitutive and facultative heterochromatin. However, I do not notice any mention about RNAi-independent pathways. Please mention such mechanisms and cite some of the relevant recent literature, e.g. Marina et al. 2013 (PMID: 24013500); Touat-Todeschini et al. 2017 (PMID: 28765164); Vo et al. 2019 (PMID: 31269446); Lee et al. 2020 (PMID:32101745).

(Response 1) Thank you for pointing this out. We added the suggested statements with citations. Because we focus on constitutive heterochromatin in this section, we did not mention facultative heterochromatin (lines 158-163):

In addition to the RNAi machinery, RNAi-independent mechanisms involving nuclear RNA processing and degradation factors such as the TRAMP (Trf4-Air1/Air2-Mtr4 polyadenylation) complex also contribute to establish heterochromatin. TRAMP containing Cid14, a member of the Trf4 family of poly(A) polymerases, targets RNAs into degradation machineries that include the exosome. Both RNAi-dependent and RNAi-independent mechanisms work in parallel to target CLRC to the repetitive DNA sequences located in the constitutive heterochromatin domains [71-77].

(Comment 2) Lines 120-132: When describe RNAi machinery please acknowledge recent high-profile papers by Roche and Martienssen 2016 (PMID: 27738016) and Folco et al. 2017 (PMID: 28199302) that underscore the role of RNAi in cell differentiation and chromosome segregation, respectively.

(Response 2) Thank you for the suggestion. We added these citations as suggested (line 113).

(Comment 3) Fig. 7: It will be useful to non-Pombe readers to have the name of the human orthologs in superscript (e.g. Amo1NUP42; Npp106NUP93).

(Response 3) Thank you for the suggestion. We modified Figure 7 to provide the name of human orthologs in superscript.

(Comment 4) Line 466: Please state “other interesting findings…” instead of “another…”. Also, the sentence is a bit awkward and may be written as “Recently, other interesting studies have demonstrated that...”

(Response 4) Thank you for pointing this out. We rewrote this statement (lines 4680469):

Recently, other interesting studies have demonstrated that ---.”